# Bibliometric Analysis of Corporate Social Responsibility and Its Impact on Community Health

**DOI:** 10.3390/ijerph22040531

**Published:** 2025-03-31

**Authors:** Mauricio Guillen-Godoy, Dennis Peralta-Gamboa, Edy Guillen-Godoy

**Affiliations:** 1School of Nursing, Faculty of Health Sciences, State University of Milagro, Milagro 091050, Ecuador; mguilleng@unemi.edu.ec; 2School of Research Training, Faculty of Research, State University of Milagro, Milagro 091050, Ecuador; 3Internal Revenue Service, Quito 170150, Ecuador; egguillen@sri.gob.ec

**Keywords:** bibliometric analysis, corporate social responsibility, community health, corporate reputation

## Abstract

This study presents a bibliometric analysis of the intersection between Corporate Social Responsibility (CSR) and community health, aiming to identify research trends, key contributors, and thematic developments in the field. Using data from Scopus, this study maps the evolution of CSR literature with a focus on public health and sustainability. The findings indicate a marked increase in publications over the past decade, with significant contributions from institutions in the United States, the United Kingdom, and China. Key themes identified include workplace health promotion, ethical concerns in corporate practices, and the role of CSR in mitigating public health risks, particularly during global crises such as the COVID-19 pandemic. This study highlights gaps in the literature and suggests directions for future research, including the need for interdisciplinary approaches and policy-driven CSR strategies. The results provide a valuable reference for academics and policymakers seeking to align corporate responsibility efforts with global health objectives.

## 1. Introduction

Corporate Social Responsibility (CSR) has emerged as an essential component of modern business management, transcending the simple obtaining of economic benefits to integrate social and environmental objectives. Several studies, including Kuakkanen y Sun [1], Anaf et al. [2] and Duon [3], highlight how companies can positively influence the health and well-being of communities through responsible practices and policies. However, the concept of CSR is broad and often overlaps with related frameworks such as Corporate Social Performance (CSP), Corporate Sustainability (CS), and Creating Shared Value (CSV).

CSR generally refers to voluntary business initiatives aimed at social and environmental improvements beyond regulatory compliance [4]. In contrast, CSP emphasizes the measurable outcomes of corporate social engagement, focusing on assessing the effectiveness and performance of CSR initiatives [5]. Corporate sustainability, on the other hand, incorporates long-term Environmental, Social, and Governance (ESG) considerations into business strategies, aligning profitability with sustainable development [6]. Lastly, Creating Shared Value (CSV) extends the CSR framework by emphasizing the integration of business and social value creation, where companies address social issues as part of their competitive advantage rather than through separate philanthropic efforts [7]. Given these conceptual differences, this study employs ‘corporate social responsibility’ and ‘CSR’ as search terms due to their widespread usage in academic literature and policy discussions. While CSR is broad, it encompasses elements from CSP, CS, and CSV, making it a suitable overarching term for capturing the diverse ways in which businesses engage with public health. Furthermore, the bibliometric approach used in this study ensures that the search results reflect various interpretations of CSR in relation to community health, allowing for a comprehensive analysis of the evolving discourse.

To understand the impact of CSR on community health, it is essential to rely on established theoretical perspectives. This study is grounded in legitimacy theory and stakeholder theory, which have been the most frequently cited in previous research on CSR and public health, according to our bibliometric analysis. These theories provide a structured framework for explaining how corporations engage in CSR initiatives that affect community well-being. Legitimacy theory suggests that companies implement CSR strategies to align with societal expectations and maintain their social license to operate [8]. Meanwhile, stakeholder theory posits that businesses are accountable not only to shareholders but also to a broad range of stakeholders, including employees, customers, policymakers, and communities [9].

CSR initiatives aimed at improving community health have gained traction as businesses increasingly acknowledge their role in addressing public health challenges. Studies indicate that CSR programs can contribute to better health outcomes by promoting access to healthcare, reducing environmental pollutants, and fostering healthier workplace environments [10,11]. The relationship between CSR and community health is particularly evident in sectors such as healthcare, pharmaceuticals, and the food industry, where corporate policies can directly impact public health outcomes. Research suggests that companies engaging in CSR activities related to public health not only improve their corporate reputation but also help to address systemic inequalities in healthcare access and social determinants of health [12,13]. Moreover, effective CSR programs can complement government policies, supporting disease prevention campaigns and public health awareness initiatives [14].

Despite growing research on CSR, there is limited systematic analysis of how it intersects with community health, particularly from a bibliometric perspective. The existing literature tends to focus on case studies or conceptual models rather than providing a structured overview of publication trends, collaboration networks, and thematic evolution in this area. This research fills that gap by conducting a comprehensive bibliometric analysis of CSR literature related to public health, identifying key contributions, thematic clusters, and influential authors and institutions.

The originality of this study lies in its systematic approach, using bibliometric methods to map the intellectual landscape of CSR and community health research. Unlike previous studies that have explored CSR in a general sense, this research offers a structured, data-driven examination of how corporate social responsibility contributes to community well-being and public health policy. This study’s significance is further underscored by the increasing role of corporations in addressing social determinants of health, particularly in the wake of global health crises such as the COVID-19 pandemic.

The findings of this study provide insights into the evolution of CSR research in relation to public health, highlight emerging areas of interest, and identify research gaps that require further exploration. These insights can guide scholars in refining future research agendas and assist policymakers in developing more effective regulations that align corporate strategies with public health objectives.

This bibliometric analysis focuses on the scientific literature on CSR and its impact on community health to identify trends, publication patterns, authors, and the most influential countries. The importance of strategic CSR management has been evidenced in the literature, showing that investing in responsible initiatives can lead to higher business profits and meet social expectations [1]. This approach includes aligning initiatives with stakeholder values, promoting social innovations, and integrating CSR into crisis management systems [15]. Furthermore, it is highlighted that CSR is not only perceived as an ethical obligation but also as a competitive advantage in today’s market [16,17]. Companies that adopt CSR practices often experience improved reputation, which can translate into increased customer loyalty and an advantage in attracting talent [18].

In the context of globalization and increasing consumer scrutiny, companies face pressures to demonstrate their commitment to sustainability and social responsibility. Effective CSR initiatives require a holistic approach that considers the economic, social, and environmental impact of business operations [15]. Furthermore, it is crucial that companies continuously assess the impact of their CSR initiatives and adjust their strategies as needed, as measurement and transparency are critical to ensure that these practices benefit the company and the well-being of society and the environment [19].

### Theoretical Framework

Understanding the relationship between Corporate Social Responsibility (CSR) and public health requires a solid theoretical foundation. This study draws on two key theoretical perspectives, legitimacy theory and stakeholder theory, which provide a comprehensive framework for explaining how corporations engage in CSR practices that impact community health.

Legitimacy theory posits that organizations seek societal approval to ensure their continued existence and success [8]. In the context of CSR and public health, businesses implement socially responsible initiatives to align with societal expectations and regulatory pressures. Companies in industries with high public health implications, such as tobacco, alcohol, and processed foods, often employ CSR strategies to maintain legitimacy despite concerns over the negative externalities of their products [20]. Through philanthropic efforts, health-related campaigns, and sustainability programs, corporations aim to mitigate public scrutiny while reinforcing their social license to operate [21].

Stakeholder theory, proposed by Freeman and Mcvea [22], expands this perspective by emphasizing that businesses are accountable not only to shareholders but also to a broad range of stakeholders, including employees, customers, policymakers, and communities. From this viewpoint, CSR initiatives related to public health emerge as a strategic response to stakeholder demands for greater corporate accountability. Companies that proactively address public health concerns—such as workplace wellness programs, community health investments, and responsible marketing practices—strengthen their relationships with key stakeholders, enhancing long-term sustainability and competitive advantage [21,23].

By integrating these theoretical perspectives, this study frames CSR as both a mechanism for corporate legitimacy and a strategic response to stakeholder expectations, highlighting how businesses navigate the intersection of corporate responsibility and public health.

## 2. Methodology

The methodology for the present study consisted of four steps: 1. Database selection; 2. Definition of terms and search equation; 3. Initial search, inclusion, and exclusion of documents; 4. Analysis of results.

Bibliometric analysis is a quantitative method used to assess the development and impact of research within a specific field by analyzing patterns in academic publications. It allows for the systematic examination of trends, author contributions, citation networks, and keyword co-occurrence to map the intellectual structure of a research domain [24]. One of the key advantages of bibliometric analysis is its ability to identify research gaps and emerging areas of study, providing valuable insights for both academics and policymakers [25]. Furthermore, bibliometric techniques enable the evaluation of international collaborations and interdisciplinary connections, offering a broader perspective on how knowledge is produced and disseminated in the field of CSR and community health.

### 2.1. Definition of Public Health and Its Impact

Public health refers to the science and practice of protecting and improving the health of populations through preventive measures, policy interventions, and community-based initiatives [26,27]. It encompasses a wide range of factors, including disease prevention, health promotion, epidemiology, and social determinants of health [28]. CSR initiatives that address public health concerns can take various forms, such as corporate-sponsored wellness programs, community-based health projects, and industry regulations designed to mitigate harmful corporate practices [29].

Building upon this definition, the impact on public health in the context of CSR refers to the ways in which corporate activities influence health outcomes at both individual and community levels. This includes improvements in health infrastructure, access to medical services, environmental health protections, and ethical business practices that reduce public health risks [8,9]. Companies that integrate public health considerations into their CSR strategies contribute to long-term community well-being and sustainability.

### 2.2. Database Selection

For the present study, the only database used was Scopus, due to its wide coverage and recognition in the academic field [24]. Different studies have used this database due to its versatility and reliability [25,30,31]. The selection of this database ensures the high quality and relevance of the publications included in the analysis.

### 2.3. Definition of Terms and Search Equation

Corporate Social Responsibility (CSR) refers to the practices and policies that companies voluntarily adopt to contribute to sustainable development [32,33]. This includes compliance with laws, ethical standards, and respect for human rights, the environment, and the community in which they operate.

Community health impact refers to the effects that the actions and policies of an entity (in this case, companies through CSR) have on the health and well-being of the population in a given community.

A search equation was developed with the selection of terms such as: “corporate social responsibility”, “CSR”, “community health”, “public health”, “community well-being”, and “collective well-being”. The resulting equation is as follows: TITLE-ABS-KEY (“corporate social responsibility” OR “CSR” AND “community health” OR “public health” OR “community well-being” OR “collective well-being”). This equation captures a wide range of papers on the topic in question, covering various dimensions related to corporate social responsibility and its impact on community health.

Book chapters and conference papers were included in this study to ensure a comprehensive analysis of the latest research developments in the field. Conference papers often present cutting-edge findings that precede journal publications, providing insight into emerging trends before they become widely accepted in peer-reviewed literature [34]. Likewise, book chapters offer in-depth discussions on specific CSR topics that might not always be covered in journal articles, contributing valuable theoretical and conceptual insights [35].

To mitigate the risk of redundancy or duplicate content, a careful screening process was implemented. Documents were reviewed to ensure that their inclusion did not overlap with subsequently published journal articles. Additionally, only conference papers from well-established and high-impact proceedings were considered, ensuring the reliability and relevance of the selected materials. This approach allows this study to capture both foundational and emerging knowledge in CSR and public health research.

### 2.4. Initial Search, Inclusion, and Exclusion of Documents

The document search was conducted on 2 July 2024, and the initial results yielded a total of 427 documents. Four criteria were used for document selection. First, documents published in the year 2024 were excluded because it was the current year. The second criterion was to include all subject areas. The third criterion was to select the following types of papers: article, review, book chapter, and conference paper. The fourth and last criterion was to choose to include all documents published in English and Spanish. After applying these criteria, 346 documents were obtained for analysis.

### 2.5. Content Analysis of the Most Influential Articles

To complement the bibliometric analysis, a qualitative content analysis was conducted to examine the key themes, methodologies, and contributions of the most influential articles in the field of Corporate Social Responsibility (CSR) and public health. Content analysis is a systematic method used in bibliometric studies to extract meaningful insights from textual data, allowing for an in-depth understanding of dominant research narratives and trends [36,37].

Methodological Approach: For the selection of influential articles, articles were ranked based on their citation impact, identified through Bibliometrix and VOSviewer. Highly cited papers were considered key contributions shaping the discourse on CSR and public health.

Categorization of Key Themes: Using thematic coding techniques [38], the content of the selected articles was analyzed to identify recurring topics such as corporate influence on public health policies, ethical business practices, sustainability commitments, workplace health initiatives, and CSR-driven responses to global health crises.

Cross-Validation of Themes: To enhance reliability, multiple researchers reviewed and coded the content independently, following established content analysis guidelines [39]. Discrepancies were resolved through discussions to ensure consistent theme identification.

Integration with Bibliometric Findings: The insights from the content analysis were triangulated with bibliometric indicators to strengthen the overall conclusions of this study. This step ensured that the qualitative synthesis complemented the network and citation-based quantitative findings.

The combination of quantitative bibliometric methods and qualitative content analysis allows for a more nuanced understanding of CSR’s role in shaping public health discussions. By systematically analyzing both citation-based impact and thematic content, this study provides a comprehensive perspective on how CSR research has evolved and what gaps remain for future exploration.

### 2.6. Analysis of Results

The bibliometric analysis was performed using VOSviewer (version 1.6.19) and Bibliometrix (R version 4.3.3), two widely accepted tools for bibliometric studies. These software packages provide robust visualization and statistical capabilities for examining research trends in CSR and public health [40,41].

Key Analytical Metrics:Publication and Citation Trends: Evaluation of the growth and impact of CSR-public health research over time.Country and Institutional Contributions: Identification of leading research nations and collaborative networks.Thematic Area Contributions: Analysis of research distribution across fields such as medicine, social sciences, and business.Keyword Mapping: Visualization of key research themes and their evolution.

A systematic approach combining bibliometric indicators with content analysis was employed to address the research question: How have research trends, publication patterns, and collaborative networks evolved in the scientific literature on Corporate Social Responsibility (CSR) and its impact on community health, and what factors influence the prominence of certain authors and countries in this field?

## 3. Results

### 3.1. Productivity and Citation Analysis

Analysis of productivity and citations over the years revealed several interesting patterns (Figure 1). In terms of publication productivity, the number of papers published has varied significantly over the years, with a notable increase in the last decade. The most productive years in terms of published papers were 2023, with 46 papers, followed by 2021 with 42 papers and 2022 with 38 papers.

On the other hand, the citations received have shown considerable variability, with peaks in certain years. The year with the highest number of citations is 2012, with 859 citations, followed by 2013 with 582 citations and 2018 with 556 citations. It is interesting to note that some years with fewer papers, such as 2012 and 2013, have a very high number of citations, which may indicate highly influential publications.

In terms of temporal trends, there is an increasing trend in recent productivity, with a notable increase in the number of papers and a variable increase in the number of citations received. This suggests a growing interest and relevance in the literature on corporate social responsibility and its impact on community health.

Descriptive statistics show that the average number of papers published per year was about 14, while the average number of citations received per year was about 263. The high standard deviation in both cases indicates large variability in the data, reflecting the different levels of influence and productivity over time (Table 1).

The first article recorded in the database dates back to 1957 and deals with the development of medical documentation in Czechoslovakia [42]. Subsequently, no publications were recorded until 1974, followed by another paper in 1976. Thereafter, there was another period of silence until 1997, when another paper was published. The year with the highest number of papers is 2023. The study with the highest impact, with 386 citations, is that by Ansari et al. [43], in which the authors evaluated how entrepreneurial initiatives can influence the improvement of social capital and capacity building in the community. The paper with the second-highest impact, with 207 citations, was published in 2013 [44]. This study revealed that the implementation of CSR practices has a significant effect on community well-being, improving quality of life and promoting sustainable and equitable development. Strategies focused on strengthening the skills and capacities of communities, in line with Amartya Sen’s theories, which are particularly effective, as they facilitate long-term empowerment and participation in socioeconomic development. Furthermore, the integration of social capital into CSR programs is crucial to ensure their sustainability and equitable distribution of benefits. Investments in public health programs, as part of CSR strategies, improve the overall health, productivity, and economic well-being of communities, especially in areas with limited access to health services. In summary, CSR practices that adopt a holistic approach, integrating capacity building and social capital, succeed in promoting sustainable and equitable development, generating significant and lasting benefits for the well-being and social cohesion of communities.

Another prominent study with 155 citations is from 2011. This article also identifies initiatives related to Corporate Social Responsibility (CSR) in the airline industry and assesses the overall state of adoption, as reported by members of the three largest airline alliances. Of 41 airlines, only 14 had publicly available annual CSR reports as of January 2009. The reports were analyzed using a qualitative content analysis approach. The results showed a greater focus on environmental issues than on the social or economic dimensions of CSR. Of the seven main environmental issues examined, emissions reduction programs predominated. Other environmental issues received much less attention, with no single initiative being implemented by all airlines. Four social and environmental issues were found, including employee well-being and engagement, diversity and social equity, community well-being, and economic prosperity. The data analysis supported arguments presented in the literature that airlines report CSR initiatives using different or inconsistent measures, making it difficult to evaluate and compare their performance and effectiveness. While a large number of airlines publishing CSR reports talked about achieving important goals (reducing emissions, increasing community involvement, or increasing workforce diversity), a much smaller number provided detailed information on specific initiatives implemented to contribute to those goals. In addition, the article raised important questions for CSR research.

The above results suggest that CSR practices that adopt a holistic approach, integrating capacity building and social capital, and succeed in promoting sustainable and equitable development, generating significant and lasting benefits in the well-being and social cohesion of communities. The assessment of the state of CSR adoption in the airline industry highlights the need for greater consistency and detail in reporting to facilitate the evaluation and comparison of initiatives and their impacts.

### 3.2. Country Contributions and International Collaborations

Figure 2 presents an analysis of scientific collaborations among 82 countries in research related to corporate social responsibility and its impact on community health. The United States tops the list with 100 non-collaborative papers, followed by the United Kingdom with 82, Australia with 27, Canada with 25, and China with 22.

On the other hand, the main research collaborator of the United States is the United Kingdom, with 11 documents, and Brazil, with 9 documents. The United Kingdom’s main collaborators were the United States and Australia, with 8 documents, and China, with 4 documents. As for Australia, in addition to the United States and the United Kingdom, its main collaborators were Switzerland (n = 3), Bangladesh (n = 2), and Canada (n = 2).

Among the U.S. publications, the study by Babor and Robaina [45] has achieved the highest impact, with 153 citations. This study provides valuable information on the Corporate Social Responsibility (CSR) initiatives of the alcohol industry and their impact on community health. It addresses issues such as the prevention of excessive alcohol consumption and education for responsible drinking. It also highlights collaboration between the alcohol industry, public health organizations, and local and national governments, presenting examples of educational campaigns and prevention programs in different countries. The study also identifies factors such as research funding, international collaborations, and the impact of national policies as determinants of companies’ activities and their impact on community health.

Another prominent study with 142 citations is that by Dofman et al. [46]. This study addresses issues such as the consumption of sugar-sweetened beverages and their relationship with obesity, highlighting how CSR campaigns of these industries have been used to divert responsibility to consumers rather than corporations. It also highlights the collaboration between these industries, public health organizations, and governments, presenting examples of initiatives such as the “Pepsi Refresh Project” and Coca-Cola’s “Live Positively”, which promote healthy lifestyles while seeking to improve corporate image. It also identifies factors such as research funding, international collaborations, and the impact of national policies as determinants in the prominence of certain authors and countries in the scientific literature on CSR and community health.

One study highlighted by the United Kingdom is the study by Herrick [47], with 90 citations. This study provides crucial information on Corporate Social Responsibility (CSR) strategies in the food industry and their relationship to obesity. This study addresses how health has become a CSR strategy, exploring issues such as preventing the consumption of unhealthy products and promoting active lifestyles. Herrick highlights the collaboration between the food industry, public health organizations, and governments, with concrete examples of programs and campaigns implemented in different countries to improve community health.

The study identifies critical factors that influence the prominence of certain actors in the field of CSR and community health, such as research funding, international collaborations, and the impact of national policies. For example, Herrick examines how companies use CSR strategies to shift responsibility for obesity from food quality to individual behavior, promoting the idea of “energy balance” and physical activity as key solutions, rather than changes in the food products themselves.

This analysis provides an in-depth understanding of the dynamics and tensions between corporate CSR initiatives and public health policies, highlighting the complexity of the relationships between the market, the state, and consumers in the context of community health.

A prominent study (n = 59 citations) from Australia is by Baum et al. [48]. This study provides a comprehensive framework for assessing the health impact of Transnational Corporations (TNCs). This study addresses the importance and methodology for conducting Corporate Health Impact Assessments (CHIAs), focusing on how the practices of these corporations affect health and equity in different global and national contexts.

The study highlights the collaboration between public health researchers, civil society activists, and TNC representatives to develop a CHIA framework that can be applied to TNC practices in various sectors, such as food and beverage and extractive industries. It also highlights the importance of factors such as research funding, international collaborations, and the impact of national policies on CHIA implementation.

The proposed CHIA framework includes assessing the global and national context of TNCs’ operations, their organizational structures, products, and business practices, and impacts on worker health and working conditions, social conditions, the environment, consumption patterns, and health-mediated economic impacts. This comprehensive approach allows for the identification of both the positive and adverse impacts of TNCs on public health, providing a sound basis for policy formulation and health promotion.

Among the collaborative works, the study by Petticrew et al. [49] stands out with 105 citations. This study, with collaboration between the UK, Switzerland, Norway, and Finland, provides a detailed analysis of how alcohol industry organizations mislead the public regarding the relationship between alcohol consumption and cancer. It addresses how the industry’s Corporate Social Responsibility (CSR) strategies distort and omit crucial information about cancer risks, identifying three main strategies used by the industry: denial/omission, distortion, and distraction. The study highlights the collaboration between alcohol industry organizations, known as ‘Social Aspect and Public Relations Organizations’ (SAPROs), and how they disseminate inaccurate or misleading information about cancer risks, denying or minimizing the link to cancers such as breast and colorectal, while emphasizing risks for less common types. In addition, it identifies factors such as research funding and national policies that influence CSR activities. This analysis provides a comprehensive view of how alcohol industry strategies affect public perception and health policies related to cancer, underscoring the need for critical evaluation of the information provided by these organizations.

Another study with collaborations from the United States, United Kingdom, Argentina, and Brazil reveals that most of the industry actions took place in high-income countries, especially in Europe and North America, and that only 27% of the actions were aligned with WHO-recommended target areas to reduce harmful drinking. In addition, most of the actions lacked scientific support, and 11% had the potential to cause harm. Benefits to the industry included brand marketing and risk management. The study concluded that alcohol industry CSR initiatives are unlikely to reduce harmful drinking, but they do provide strategic commercial advantages while appearing to have a public health purpose [50].

### 3.3. Analysis of the Contributions of the Main Thematic Areas

The three areas with the largest contributions are Medicine, with 172 papers; Social Sciences, with 124 papers; and Business and Management, with 70 papers (Figure 3). Table 2 shows the top three authors with the most contributions in these five subject areas and the number of papers they contributed.

Medicine accounts for 28.2% of the documents. The first document appeared in 1967, followed by a silence until 1974, and then another until 1997, with only one document in each of these years. The highest number of documents was recorded in 2022, with 23.

An article in this area analyzes the chemical industry’s “Responsible Care” Corporate Social Responsibility (CSR) campaign, initiated in 1988. Although advocates of CSR argue that corporations should address the needs of all stakeholders, not just shareholders, the article suggests that the main goal of responsible care has been to change public perception and oppose stricter public health and environmental legislation. This analysis highlights the evolution of CSR practices in the chemical industry, the use of strategic publications to influence public opinion and policy, and the interaction between industry, the public, and legislators. It also highlights how these dynamics can influence the prominence of certain authors and countries in the CSR literature. This critical approach provides a holistic view of how CSR initiatives can serve both commercial objectives and public interests [51].

Another study by Gómez et al. [12] examines how the sugar-sweetened beverage industry uses CSR programs to improve its image and exert political influence, despite the negative health effects of its products. It shows how CSR strategies have evolved in response to scientific evidence and public health policies. It highlights publication patterns that favor industry interests and strategic collaborations between companies, governments, and public health organizations. In addition, it illustrates how funding and support for specific programs can increase the prominence of certain authors and countries in the scientific literature on CSR, often to the detriment of public health.

Millar [13] indicates that corporations negatively influence health, underscoring the need for urgent government action. Kumar et al. [52] examine how the tobacco industry interferes with control policies and collaboration between government and civil society. Another study analyzes how the German media debate how the sugar tax frames social and governmental responsibility.

Social Sciences contributed 20.3% of the documents. The first Social Sciences papers appeared in 2004, reaching the highest number of publications (n = 18) in 2023.

The study by Pantani et al. [53] analyzes the German media debate on sugar taxation, highlighting the concepts of social and governmental responsibility. Through an analysis of 114 German national newspaper articles, the narrative frames used in the sugar tax discussion are identified and examined. The results show a conflict between individual and social responsibility, with a focus on how to fulfill social responsibility through legislative measures versus voluntary industry commitments.

Fooks et al. [44] examines the boundaries of Corporate Social Responsibility (CSR) using internal tobacco industry documents to explore how British–American Tobacco (BAT) has used CSR to influence policy and avoid stricter regulations. The study presents a three-stage model based on Sykes and Matza’s theory of neutralization techniques, which details how BAT executives justify their actions and manipulate public and legislators’ perceptions. The analysis underscores the importance of viewing managers’ public statements critically and placing them in their economic, political, and historical contexts.

Leone et al. [54] explore how UK companies use CSR to promote physical activity in children, in the context of the Public Health Responsibility Deal program. They compare CSR strategies across different industry sectors and analyze whether these initiatives effectively support children’s right to play and be physically active. The results show that while many companies are committed to promoting physical activity, these strategies are often used as compensatory tactics to divert attention from health problems related to their core products, such as childhood obesity.

Monachino and Moreira [14] investigate how CSR in the pharmaceutical industry contributes to global health governance, highlighting multi-sectoral partnerships and health policies.

Moggi et al. [55] analyze how CSR at farmers’ markets reduces food waste and improves public health through collaborative networks.

Business and Management contributed 11.5% of the documents. Studies such as the one by Flanagan and Whiteman [56] demonstrate that corporate social responsibility can be seen in practice as a dynamic negotiation and interaction between multiple actors. They were able to negotiate price reductions for HIV drugs and develop local production capacity, thus avoiding a public health disaster.

Another study in this thematic area explores how Corporate Social Responsibility (CSR) can influence societal happiness. It argues that corporations have a social responsibility to respect, preserve, and promote people’s happiness. Chia et al. [57] propose a comprehensive conceptual framework that includes the macro-to-micro and micro-to-macro pathways through which corporate activities directly and indirectly impact societal happiness. The study highlights the need for a holistic conceptualization of happiness that encompasses objective, subjective, hedonic, and eudaimonic dimensions. Furthermore, it underscores the importance of adequately measuring happiness and considering the impacts of corporate activities beyond financial performance, advocating that companies should be oriented towards promoting well-being and happiness rather than mere wealth accumulation.

The study by Lee [58] explores the implications of CSR in the tourism and hospitality industry during the COVID-19 pandemic. It analyzes how companies in the industry adapted their CSR initiatives to cope with the crisis, highlighting the importance of CSR strategies from both a financial and strategic management perspective. The study discusses the evolution of CSR in response to the pandemic, the financial impact of these initiatives, and the relevance of culture and industry as boundary conditions. It also examines company performance measures and the role of CSR in risk management during and after the pandemic.

### 3.4. Keyword Mapping

To examine the intellectual structure of the field of study, a co-occurrence analysis of the author’s keywords is required. This analysis consists of constructing a network of terms (author’s keywords) that appear most frequently in the literature [59]. In generating this network, a total of 1037 keywords were identified, and by prioritizing a minimum number of occurrences of occurrences (n = 5), 28 keywords were observed (Figure 4). The keyword with the highest frequency of occurrence was “financial literacy” (n = 31 occurrences), followed by “financial inclusion” (n = 8 occurrences). In third place were “small business” and “financial education”, with 7 and 4 occurrences, respectively.

The trend observed between 2014 and 2016 (Figure 4), which includes terms such as “community health”, “sustainable development”, “policy”, “conflict of interest”, “health promotion” and “tobacco”, suggests a growing interest in the impact of Corporate Social Responsibility (CSR) policies on various aspects of public health and sustainable development. This trend reflects how companies are increasingly committed to initiatives that not only improve their corporate image but also contribute to community health promotion, sustainable development, and transparency in their operations.

This trend reflects a focus on improving the health and well-being of communities, a key priority in CSR strategies, For example, the study by Henke et al. [60] indicates that investments in a company’s internal health culture predict improvements in some employee health risks and healthcare utilization. Sustainable development indicates a commitment to sustainable practices that balance economic, social, and environmental needs [61,62,63].

This trend suggests increasing attention to the formulation and implementation of policies that support CSR and its objectives. The study by Green et al. [64] highlights the increasing adoption of policies by large e-commerce platforms that restrict legal wildlife trade. These policies seek to mitigate the harmful effects of commercial wildlife trade on global biodiversity and public health. Likewise, policies that support innovation and preventive practices in dental care are addressed [65]. On the other hand, the study by Wang et al. [66] demonstrates that mandatory Corporate Environmental Responsibility (CER) disclosure policies can significantly reduce environmental pollution. This highlights the importance of implementing policies that force companies to disclose environmental information to promote CSR. This trend underscores the relevance of CSR as an integral tool for addressing public health issues and promoting more equitable and sustainable development.

In the period of 2017 to 2018, the change in keywords with more occurrences is evident; the most-used words were “corporate social responsibility”, “public health”, “obesity”, “tobacco industry”, “alcohol industry”, “ethics”, “sustainability”, “gambling”, “alcohol”, and “tobacco control”. These words suggest a growing interest in the intersection of corporate social responsibility with public health and ethical issues, as well as in the regulation of industries that significantly impact the health and well-being of society, such as the tobacco, alcohol, and gambling industries. This shift reflects a greater concern for sustainability and the ethical impact of corporate practices on public health.

In this trend, obesity is presented as a public health problem and its connection with the food and beverage industry [67,68]. There is evidence of a trend in publications such as gambling companies, alcohol [69], and tobacco [70] and their importance in the corporate responsibility industry [71]. In addition, the trend emphasizes the ethics to be followed by companies in their corporate responsibility activities, as the study by Poswa and Davies [72], which addresses ethics in business practices and Corporate Social Responsibility (CSR) activities primarily through the assessment of ethical practices and standards in the South African mining industry, with a particular focus on mine tailings management. It examines how mining companies deal with ethical challenges and regulatory compliance, and how they integrate CSR into their operations. The text analyzes the effectiveness of current legislation and concludes that it is largely ineffective, with a low level of adherence by mine management and the mining community. Weaknesses in compliance monitoring mechanisms are also highlighted. New perspectives on legislative issues for unresolved problems in tailings management are presented, and directions for future research are indicated. In addition, the study by Poswa and Davies [72] discusses how companies use CSR as a political management strategy to neutralize the impact of external social actors and minimize political support for regulatory changes rather than genuinely addressing important ethical issues.

Starting in 2019, there was a shift in keywords with increased frequency, with “workplace promotion”, “food industry”, “commercial determinants of health”, “China”, and “COVID-19” standing out. These words suggest a growing interest in the intersection between public health and corporate practices, especially concerning workplace health promotion, the influence of the food industry on health, and commercial determinants of health. In addition, the emergence of “China” and “COVID-19” highlights the attention to global health and the impacts of the pandemic on business and public health practices worldwide. This shift reflects an adaptation of research and policy agendas to address emerging health challenges and the influence of commercial factors on population health. In this context, CSR improves organizational resilience, satisfaction, trust, and post-crisis recovery [73]. Also, the study by Valls Martinez et al. [74] indicates that COVID-19 negatively affected the economic sustainability of private hospitals in Spain, highlighting the importance of public spending on health and corporate social responsibility to maintain profitability during the crisis.

## 4. Discussion

Corporate Social Responsibility (CSR) has emerged as an essential component of modern business management, transcending the simple obtaining of economic benefits to integrate social and environmental objectives. Several studies, such as those by Kuokkanen and Sun [1] and Duong [3], highlight how companies can positively influence the health and well-being of communities through responsible practices and policies. CSR initiatives aimed at improving community health have gained traction as businesses increasingly acknowledge their role in addressing public health challenges. Studies indicate that CSR programs can contribute to better health outcomes by promoting access to healthcare, reducing environmental pollutants, and fostering healthier workplace environments [8,9].

The relationship between CSR and community health is particularly evident in sectors such as healthcare, pharmaceuticals, and the food industry, where corporate policies can directly impact public health outcomes. Research suggests that companies engaging in CSR activities related to public health not only improve their corporate reputation but also help to address systemic inequalities in healthcare access and social determinants of health [10,11]. Moreover, effective CSR programs can complement government policies, supporting disease prevention campaigns and public health awareness initiatives [14].

This bibliometric analysis focuses on the scientific literature on CSR and its impact on community health to identify trends, publication patterns, authors, and the most influential countries.

The importance of strategic CSR management has been evidenced in the literature, showing that investing in responsible initiatives can lead to higher business profits and meet social expectations [1]. This approach includes aligning initiatives with stakeholder values, promoting social innovations, and integrating CSR into crisis management systems [15]. Furthermore, it is highlighted that CSR is not only perceived as an ethical obligation but also as a competitive advantage in today’s market [14,15]. Companies that adopt CSR practices often improve their reputation, which translates into increased customer loyalty and an advantage in attracting talent [18].

### 4.1. Implications for the Body of Knowledge

The growing integration of CSR with public health demonstrates a paradigm shift in corporate responsibility, where businesses are increasingly expected to take proactive roles in addressing societal health challenges. The alignment of CSR with global sustainability goals highlights the need for interdisciplinary research, bridging business ethics, public health policy, and environmental science. This shift reinforces the role of CSR in tackling social determinants of health, such as healthcare access, environmental sustainability, and workplace well-being.

One of the most notable contributions to the academic body of knowledge is the recognition that CSR is no longer an isolated corporate function but a strategic tool embedded in governance and policy frameworks. Studies increasingly emphasize that CSR initiatives must move beyond voluntary commitments and be subject to regulatory compliance, performance evaluation, and long-term impact assessment. This necessitates further research into standardized methodologies to evaluate the effectiveness of CSR programs in improving public health outcomes.

### 4.2. Identified Trends

During 2014–2016, the most frequent keywords in the literature included “community health”, “sustainable development”, “policy”, “conflict of interest”, “health promotion”, and “tobacco industry”. These trends reflect a growing interest in the impact of CSR policies on various aspects of public health and sustainable development. Companies began to focus on how their practices could contribute to health promotion in the community, balancing economic, social, and environmental needs [75,76]. This period coincides with the increasing pressure from regulatory bodies and non-governmental organizations advocating for stronger corporate commitments to sustainability and health. Governments in various regions started implementing stricter regulations regarding corporate disclosures, prompting firms to integrate CSR into their business strategies.

Between 2017 and 2018, the most frequently used keywords were “corporate social responsibility”, “public health”, “obesity”, “tobacco industry”, “alcohol industry”, “ethics”, “sustainability”, “gambling”, and “tobacco control”. This shift indicates an increased interest in the intersection of CSR with issues of public health and ethics, as well as in the regulation of industries that have a significant impact on the health and well-being of society, such as the tobacco, alcohol, and gambling industries. The growing societal concern regarding the long-term health consequences of these industries led to intensified lobbying for stricter policies, taxation, and marketing restrictions. Companies in these sectors started employing CSR strategies as a means to counteract negative perceptions, often funding public health initiatives while continuing to engage in controversial practices. This period also saw the emergence of investor activism, where shareholders and stakeholders began demanding accountability from corporations regarding their impact on public health [72,77,78].

Starting in 2019, there was a shift toward terms such as “workplace promotion”, “food industry”, “commercial determinants of health”, “China”, and “COVID-19”. These words suggest a growing interest in the intersection between public health and corporate practices, especially in relation to workplace health promotion and the influence of the food industry on health. The emergence of “COVID-19” highlights the attention to global health and the impacts of the pandemic on business and public health practices. The pandemic accelerated the need for companies to integrate CSR as a strategic tool, not only to enhance corporate reputation but also to ensure business resilience in times of crisis. Many firms shifted their CSR efforts to pandemic relief, supporting employees, healthcare infrastructure, and vulnerable populations, reinforcing CSR’s critical role in crisis management. This period also witnessed a transformation in consumer expectations, as customers increasingly favored companies that demonstrated social responsibility through direct action rather than symbolic gestures. As a result, industries that were traditionally less involved in health-related CSR, such as the technology and finance sectors, began engaging in initiatives that supported public health and pandemic recovery efforts [79]. This shift reflects an adaptation of research and policy agendas to address emerging health challenges and the influence of commercial factors on population health [80].

CSR has become a fundamental component for companies aiming to enhance their reputation, sustainability, and make a positive impact on public health. The trends identified in this bibliometric analysis indicate an evolution towards a more strategic CSR aligned with global sustainability and public health, with an increasing focus on emerging issues such as the COVID-19 pandemic and workplace health promotion. Furthermore, the integration of CSR with policies addressing economic and social inequalities is becoming more prominent, suggesting a growing expectation for companies to actively contribute to broader societal well-being beyond their immediate stakeholders. The evolution of CSR reflects the increasing interdependence between corporate activities and societal health, making it imperative for companies to proactively address public health concerns rather than reactively managing crises. This shift underscores the role of CSR as a tool for long-term sustainability and social impact. Companies must continue to adapt their CSR strategies to address these challenges and maximize their contribution to the well-being of the communities in which they operate.

### 4.3. Synthesis of Key Trends in the Literature

Rather than focusing on individual studies with high citation counts, this study critically synthesizes the broader trends observed in the literature on CSR and community health. The findings indicate that CSR has evolved from a corporate reputation strategy to a multifaceted approach that integrates health, sustainability, and stakeholder engagement. This transformation is evident in the increasing emphasis on workplace health promotion, ethical corporate governance, and the intersection between CSR and regulatory frameworks [81,82].

Over time, CSR has transitioned from a voluntary commitment by corporations to a more structured and strategic tool aligned with public health and sustainability. Early CSR initiatives focused primarily on philanthropy and corporate image, whereas recent approaches highlight proactive engagement in health promotion, environmental sustainability, and responsible business practices.

#### 4.3.1. Key Transformations 

Workplace Health Promotion: Companies have increasingly invested in employee well-being programs, addressing occupational health, mental health awareness, and preventive care initiatives.

Ethical Corporate Governance: A growing body of research emphasizes the role of ethical leadership and transparent business operations in fostering public trust and mitigating corporate malpractices affecting health.

Regulatory Compliance and Policy Integration: CSR initiatives are now more closely linked to regulatory frameworks, ensuring alignment with national and international health standards.

#### 4.3.2. Integration with Global Sustainability Goals

In particular, research trends suggest that CSR initiatives addressing public health concerns have become increasingly aligned with global sustainability objectives [83].

Over the past decade, studies have highlighted corporate efforts to mitigate health risks through various strategies, including:Reducing Carbon Footprints: Many companies have implemented sustainability programs to minimize environmental impact, recognizing the direct link between pollution and community health.Ethical Sourcing and Responsible Supply Chains: There is a growing emphasis on ethical sourcing practices, particularly in the food and beverage industry, to reduce health risks associated with harmful additives and unsustainable agricultural practices.Community-Based Wellness Initiatives: Organizations are investing in health education, disease prevention campaigns, and access to medical services in underserved communities.

These developments reflect a shift toward CSR strategies that extend beyond philanthropy and actively contribute to public health policies and long-term community well-being. The increasing interdependence between corporate activities and public health underscores the need for businesses to integrate health considerations into their strategic planning. Future research should further explore the role of CSR in bridging economic objectives with social impact, ensuring that corporate responsibility efforts lead to tangible health improvements at both local and global levels.

## 5. Conclusions

This study serves as a valuable benchmark for future research by providing a comprehensive overview of the evolution of CSR literature and its implications for community health. By mapping the trends, key contributions, and international collaborations in this field, this research establishes a foundation for scholars seeking to explore new dimensions of CSR’s impact. Furthermore, the methodological approach employed, combining bibliometric analysis with thematic mapping, offers a replicable framework for similar studies in other industries and contexts. This work not only enhances understanding of the current state of CSR research but also guides policymakers and business leaders in aligning corporate initiatives with sustainable development goals.

### 5.1. Theoretical Contributions

This study contributes to the theoretical understanding of CSR by reinforcing the role of legitimacy theory and stakeholder theory in explaining corporate motivations for engaging in socially responsible practices that influence community health. The findings confirm that businesses increasingly implement CSR strategies to enhance legitimacy and meet stakeholder expectations, particularly in industries with significant public health implications. Additionally, this research highlights the shift from traditional philanthropic CSR initiatives to more strategic, integrated approaches that align with sustainability and regulatory compliance.

### 5.2. Practical Implications

From a practical perspective, this study provides insights for corporate decision-makers on how CSR initiatives can be effectively designed to contribute to public health while fostering business sustainability. Companies should focus on integrating health-related CSR initiatives into their core business strategies, ensuring that these efforts go beyond superficial branding and contribute to long-term community well-being. Furthermore, policymakers can leverage these findings to develop regulatory frameworks that encourage corporations to engage in meaningful and transparent CSR activities, particularly in industries such as food, pharmaceuticals, and environmental health.

### 5.3. Practical Recommendations

Based on the findings of this study, several practical recommendations emerge to guide future CSR practices and policy interventions:Strengthen CSR-Driven Public Health Policies: Policymakers should develop frameworks that incentivize businesses to implement CSR initiatives aligned with public health priorities. These policies should include clearer accountability mechanisms to ensure meaningful contributions rather than superficial branding.Develop Standardized Metrics for CSR Impact Assessment: To improve transparency, there is a need to create globally recognized metrics for measuring the actual health benefits of CSR initiatives. This will allow businesses, researchers, and regulators to track long-term outcomes effectively.Encourage Industry-Specific CSR Strategies: Different industries require tailored CSR approaches. Companies in sectors such as pharmaceuticals, food production, and manufacturing should align their CSR initiatives with public health concerns unique to their field.Promote Cross-Sector Collaboration: Governments, corporations, and civil society organizations should work together to create multi-stakeholder initiatives that leverage CSR for health promotion. Public–private partnerships can enhance the scalability and effectiveness of CSR-driven programs.Enhance Transparency and Consumer Awareness: Companies should prioritize transparent reporting on their CSR initiatives, ensuring that consumers can make informed choices. Third-party certifications and independent audits can reinforce credibility and accountability in CSR efforts.

### 5.4. Identified Gaps in the Literature

Despite the growing body of research on CSR and community health, several gaps remain:Measurement of CSR Impact: There is a lack of standardized methods to quantify the tangible health benefits of CSR initiatives, making it difficult to assess their effectiveness.Longitudinal Studies: Most studies focus on short-term CSR impacts, with limited research on how these initiatives contribute to sustainable community health over time.Geographical and Industry-Specific Insights: Research is disproportionately concentrated in developed economies, with fewer studies examining CSR’s role in public health within emerging markets and specific industries such as fast fashion, technology, and agriculture.

### 5.5. Recommendations for Future Research

Future research could explore:The Long-Term Effects of CSR Initiatives on Community Health—Assessing sustainability and real impact beyond corporate branding.The Role of Regulatory Frameworks in Shaping CSR Practices—Particularly in industries with high public health implications such as alcohol, tobacco, and processed foods.CSR Strategies in Emerging Economies—Understanding how economic development, governance, and cultural expectations shape corporate responsibility dynamics.Interdisciplinary Approaches—Integrating insights from public health, economics, and business ethics to develop a more comprehensive framework for assessing CSR’s contribution to sustainable development and social well-being.

By addressing these research gaps, future studies can build on this work to refine CSR strategies, enhance their effectiveness, and contribute to the broader discourse on corporate responsibility and public health. Strengthening the link between CSR and tangible health outcomes will be crucial for advancing both academic knowledge and practical applications in the field.

## Figures and Tables

**Figure 1 ijerph-22-00531-f001:**
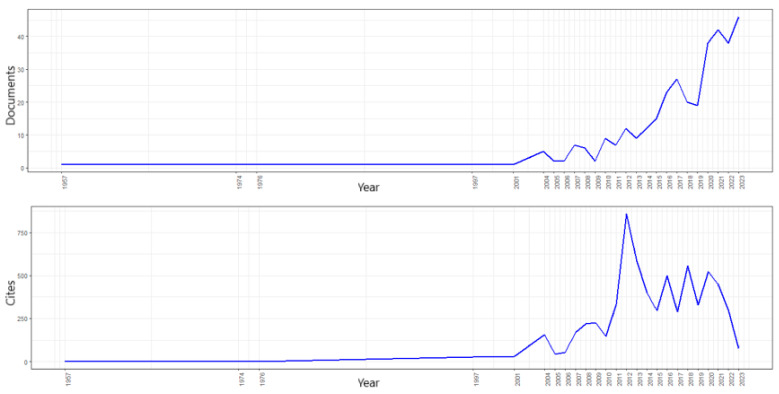
Documents and citations by year.

**Figure 2 ijerph-22-00531-f002:**
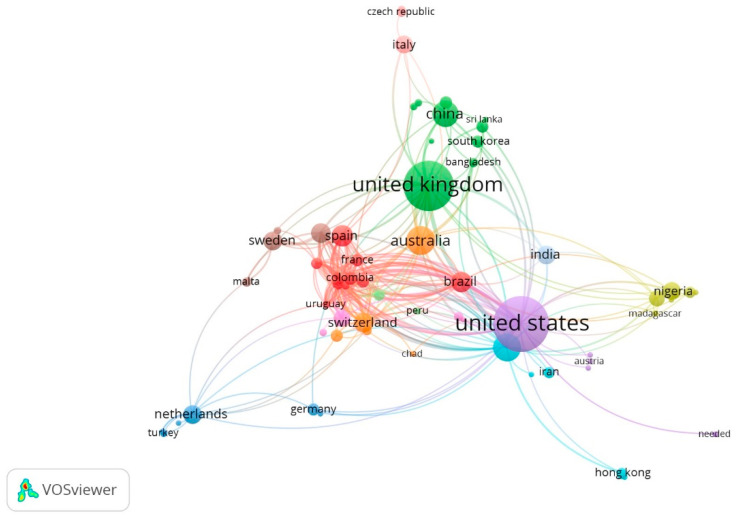
International collaborations.

**Figure 3 ijerph-22-00531-f003:**
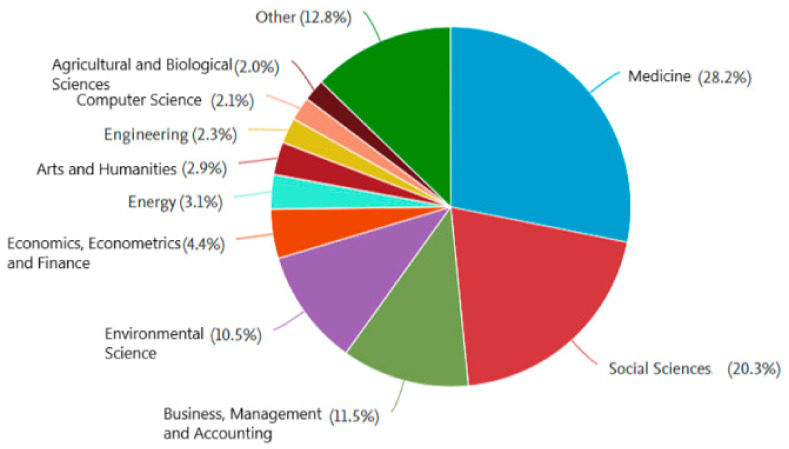
Documents by subject area.

**Figure 4 ijerph-22-00531-f004:**
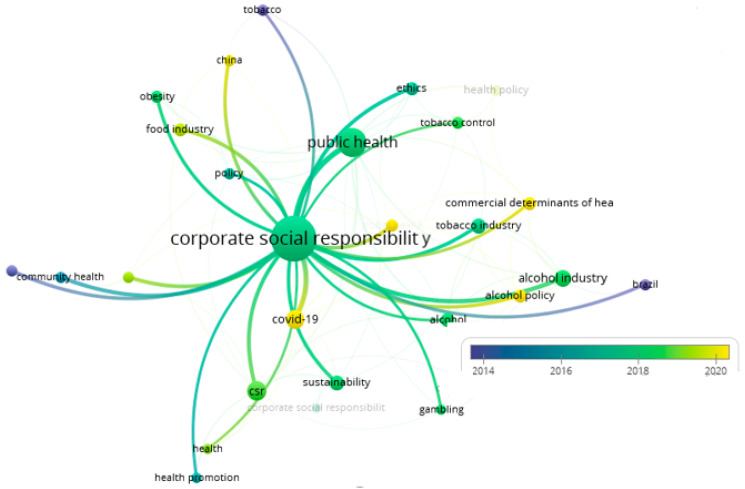
Keyword co-occurrences over time.

**Table 1 ijerph-22-00531-t001:** Academic production and citation statistics.

Element	Detail
Most productive year (documents)	2023 (46 documents)
Year with more citations	2012 (859 citations)
Average number of documents per year	13.84
Average number of citations per year	263.08
Standard deviation (documents)	14.21
Standard deviation (citations)	224.65

**Table 2 ijerph-22-00531-t002:** Top 3 authors in the 3 main thematic areas.

Thematic Area	Name	Country	Institution	Documents
Medicine	Petticrew, M.	United Kingdom	London School of Hygiene & Tropical Medicine	8
	McCambridge, J.	United Kingdom	University of York	6
	Macassa, G.	Sweden	Hogskolan i Gavle	5
Social Sciences	Macassa, G.	Sweden	Hogskolan i Gavle	5
	Benson, P.	Sweden	Hogskolan i Gavle	3
	Collin, J.	United Kingdom	University of Edinburgh	3
Business and Management	Hastings, G.	United Kingdom	University of Stirling	2
	Kim, Y.	United States	University of Texas at Austin	2
	Agatiello, O.R.	Switzerland	Geneva School of Diplomacy and International Relations	1

## Data Availability

The data used in this study are available in the Scopus database. The collection was performed using the following search equation: TITLE-ABS-KEY (“corporate social responsibility” OR “CSR” AND “community health” OR “public health” OR “community well-being” OR “collective well-being”). The inclusion and exclusion criteria described in Section 2 were applied.

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
