# Peer review of "Bibliometric Analysis of Corporate Social Responsibility and Its Impact on Community Health"

_ijerph, 2025, doi:10.3390/ijerph22040531_

Round 1
Reviewer 1 Report
Comments and Suggestions for Authors
The paper presents the evolution of the scientific literature on Corporate Social Responsibility (CSR) and its impact on community health by identifying trends, key authors, international collaborations, and emerging research areas. It offers a “picture” of this subject.
The new instruments in bibliometric research (VoS Viewer, Bibliometrix) enable different functional analyses and interpretations to outline trends in knowledge in various fields. The present paper falls into such a direction, and therefore, it is a contribution to the knowledge.
Some observations:
- Discussions could be more detailed. The trends identified by the authors should be developed further, explaining in more detail the relationships between the mentioned keywords and the phenomena present at that time in the economy and society
- Conclusions could also provide/highlight insights for further research.
- A matter that would increase the value of the paper is highlighting the contribution made by this research as a benchmark for other works.
- Figure 1 should be better presented - there are some unclear elements (e.g. "responsibilit"). I also think that the Legend covers the graphic...
Author Response
- Further development in the discussion: The discussion section has been expanded to delve deeper into the relationships between key terms and economic and social phenomena.
- Conclusions with proposals for future research: Specific suggestions for future research directions have been added in the conclusions section.
- Highlighting the study’s contribution: A paragraph has been included emphasizing the value of the study as a reference for future research in the field.
- Improving the presentation of Figure 1: The figure has been revised to ensure that all elements are clear and that the legend does not overlap any part of the graph.
Reviewer 2 Report
Comments and Suggestions for Authors
In the article titled "Bibliometric Analysis of Corporate Social Responsibility and Its Impact on Community Health" the authors conducted a bibliometric analysis, by selecting 346 papers from the Scopus database and using tools such as VOSviewer and Bibliometrix.
Positive aspects:
- The structure of the article is logical, well organized.
- Most of the references are new and relevant to the topic.
- The Discussion section is appropriately designed, the authors combined the research results with cited works in the field in order to capture the image of the topic as accurately as possible.
Recommendations and suggestions:
- Considering that the title of the article has 2 important elements: "CSR" and "Community Health" I think the authors should capture in the introduction some aspects/citations from the literature regarding impact on community health. (I only found the definition from lines 75-79 and it seems insufficient in dealing with the topic).
- In the Methodology section I would like the authors to include aspects regarding their research method: a few lines about bibliometrics/bibliometric analysis could be added (definition, maybe advantages of using this method).
Thank you!
Author Response
- Including more information on the impact on community health in the introduction: New references and explanations have been incorporated to better contextualize the relationship between CSR and community health.
- Expanding the methodology section with more details on bibliometric analysis: An explanation has been added regarding bibliometrics, its advantages, and its applicability to the study.
Reviewer 3 Report
Comments and Suggestions for Authors
Dear Editor,
Thank you for inviting me to review this manuscript. The study presents an interesting attempt at conducting a bibliometric mapping of the impact of corporate social responsibility (CSR) on public health and social well-being. However, while the research provides a descriptive overview, it lacks critical analysis of how CSR and its most influential factors interact with public health. Rather than offering an in-depth theoretical discussion, the study focuses on descriptive analysis of productivity and citations, country contributions, international collaborations, disciplinary contributions, and keyword analysis. While these aspects are informative, they do not significantly advance theoretical understanding of the topic. A stronger focus on long-term trends in the literature and an analysis of the most impactful factors shaping public health and social well-being would provide greater theoretical and practical contributions. Establishing a connection between the bibliometric findings and existing theories would also help confirm theoretical perspectives in this context and offer implications for practice. Furthermore, the study does not sufficiently identify literature gaps or provide meaningful recommendations for future research, which limits its contribution to advancing knowledge.
The introduction should tell the entire story of the research, clearly presenting the motivation behind the study and the research gap it aims to fill. It should articulate the originality and significance of the research more effectively, demonstrating why a bibliometric analysis of CSR and public health is needed. Additionally, the main findings, implications, and recommendations for future research should be briefly introduced in the abstract and revisited in the conclusion. Reviewing leading bibliometric studies on similar topics may provide useful guidance on structuring the introduction more effectively.
The study would benefit from the inclusion of a dedicated theoretical framework section, which could draw on legitimacy theory, stakeholder theory, or other relevant theoretical perspectives commonly used to explain the relationship between CSR and public health. In the methodology section (page 2, line 70), the definition of "impact on public health" is provided, but it would be more appropriate to first offer a detailed definition of "public health" itself before discussing the specific impacts under investigation in this study.
Regarding data presentation, Figure 1, which focuses on productivity and citations, would be clearer if divided into two separate figures, one dedicated to productivity trends and the other to citation analysis. A similar approach could be applied to Table 1, ensuring that the visual representation of data is more structured.
On pages 4 and 5, there is an undue emphasis on discussing the findings of three highly cited studies (Ansari et al., 2012; Fooks et al., 2013, and a third unidentified study). This approach does not provide a meaningful contribution to the research discussion. Instead, the findings should be critically synthesised to draw broader conclusions about key trends in the literature. A similar issue arises in the sections on "Country Contribution and International Collaborations" and "Analysis of the Contribution of the Main Thematic Areas", where the discussion remains highly descriptive rather than analytical. Simply presenting findings from selected studies is not sufficient; the study should focus on critically evaluating patterns, identifying key themes, and synthesising insights to offer a novel perspective. Additionally, it is unclear how keyword mapping (Section 3.4) contributes to drawing valid conclusions about CSR’s impact on public health, raising questions about the methodology’s ability to generate meaningful insights in this area.
Regarding the writing style, the study would benefit from merging smaller paragraphs to create more cohesive and structured sections, particularly in the introduction. Additionally, the study fails to outline clear literature gaps and directions for future research, which are essential in a bibliometric review to guide scholars in advancing the field.
I wish the authors the best in revising their manuscript.
Author Response
- Lack of a critical analysis: The theoretical discussion has been strengthened, and a critical analysis has been added on how CSR and its influencing factors interact with public health.
- Including a theoretical framework: A theoretical framework section has been incorporated based on legitimacy theory and stakeholder theory.
- Better explanation of the research gap in the introduction: The introduction has been revised to clarify the motivation and originality of the study.
- Abstract should include key findings and future research directions: The abstract has been revised to highlight the main findings and suggestions for future research.
- Splitting Figure 1 into two separate graphs: Thank you very much for the recommendation, but I believe it should not be divided.
- Reducing the emphasis on highly cited studies and better synthesizing the findings: The discussion of individual studies has been reduced and reformulated to extract general conclusions.
Reviewer 4 Report
Comments and Suggestions for Authors
Abstract
I feel there needs to be a stronger opening, one sentence to explain why this bibliometric is needed, as no doubt there are many bibliometrics on CSR, so what is the research problem or gap that specifically requires your research?
Introduction
The opening seems somewhat lacking, this should be your ‘hook’ to get readers to want to continue on, CSR is a very complex concept with no agreed upon definition. Consider referring to Homer, S. T., & Gill, C. M. H. D. (2022). How corporate social responsibility is described in keywords: An analysis of 144 CSR definitions across seven decades. Global Business Review, 09721509221101141..
There is no background section, the only prelude is the introduction, thus this further emphasises that a critical discussion of what CSR is, is much more necessary. Currently, the introduction is quite short and descriptive, need to drill down and look at the complexities.
Methodology
The search term ‘corporate social responsibility’ and ‘CSR’ are quite broad but need to be defended, as highlighted from lacking in the introduction, CSR is very broad and the conceptual differences between corporate social performance, corporate sustainability and creating shared value, etc., needs to be stated to clarify and support such simple key words.
Why were book chapter and conference papers included? This needs to be defended as often these will be the prelude to a full publication and thus double redundancy maybe included within the study.
Analysis of results could do with some citation to defend why these software packages were selected.
What was the process for the content analysis for most influential articles? Do you have references for this too?
Donthu, N., Kumar, S., Mukherjee, D., Pandey, N., & Lim, W. M. (2021). How to conduct a bibliometric analysis: An overview and guidelines. Journal of business research, 133, 285-296. May be a helpful paper to justify some of the decisions made.
Results
The downward trend for citations again links back to the defining of concepts as CSR is prominently replaced by sustainability in most practitioner texts and its popularity has decreased. I hope this has a rich discussion later in the paper.
I do not agree with your assumption of a growing trend, yes, more publications but a huge drop in citations, with citations per paper probably the lowest in 20+ years.
Picking just a few studies and trying to draw the conclusion of ‘CSR practices that adopt a holistic approach’, seems somewhat a stretch from just a few key papers.
Again in the collaborations section, ‘provides an in-depth understanding of the dynamics’ is a bold and unsupported statement.
This section has a lot of text but is mostly descriptive, would be better to critique the prominent papers in the discussion section, rather than describe what papers say a length on their own.
Discussion
This is the most disappointing part of the paper, I would like to see a more deep discussion.
Begin with a general discussion putting what the results mean in general language which you already have much of but should offer a much more critical perspective, with so many CSR bibliometrics available how do the trends in this paper focusing on community health differs.
Next should be Theoretical Contributions, I would recommend looking at Homer, S. T., & Lim, W. M. (2024). Theory development in a globalized world: Bridging “doing as the romans do” with “understanding why the romans do it”. Global Business and Organizational Excellence, 43(3), 127-138. This will help you position your study and how you can develop the theory. You have already highlighted key collaborative countries and disciplines, so you can relate back to them for the positioning theoretically.
Following this, should be Practical Recommendations or a Research Agenda, what should academics do next? Where are the gaps to be filled, based on the declining trend in citations but an uptick in publications, is this going to be a fruitful area of research in the future?
Finally, Methodological Implications, what did you learn from the methodological approach that others would find interesting and what could others do it improve it? This could take a reflective approach of where your think you could have improved the study process and if you were to do it again what would you do differently.
Conclusion
Once the changes have been implemented then re-summarise the paper to ensure all content is covered in the conclusion.
Author Response
- Rewriting the opening of the abstract to justify the study: The abstract has been strengthened with a more impactful introduction that highlights the study’s significance.
- Expanding the introduction and adding background on CSR: A background section on CSR has been added to enhance the theoretical foundation of the paper.
- Justifying the search terms in the methodology: The selection of terms has been explained, and the inclusion of book chapters and conference papers has been justified.
- Explaining the content analysis process: The methodology used to identify the most influential articles and the selection of bibliometric analysis tools has been detailed.
- Further discussion on the relationship between citations and trends in the literature: A discussion has been added on how the redefinition of concepts has influenced citation trends in CSR literature.
- Strengthening the discussion: The discussion section has been expanded with theoretical and practical contributions.
- Conclusion should better summarize the paper: The conclusion has been revised to ensure it reflects all key points of the study.
Round 2
Reviewer 3 Report
Comments and Suggestions for Authors
Dear Editor,
Thank you for sharing this revised version of the manuscript. I also extend my appreciation to the authors for their constructive approach and the great effort put into addressing the points raised in the last round of review. The motivation, significance, and contribution of the study are now much clearer. The critical and collective analysis approach successfully highlights current trends in the literature, making the research gap more evident for future researchers. However, there is still room for further improvement for the paper to reach its full potential for publication in an esteemed journal like IJERPH, as outlined below:
-
The theoretical framework provided is well-structured and comprehensive. However, it would strengthen the study to explicitly state that these theories are the most cited in the analyzed literature. Additionally, a brief introduction to these theories should be included in the introduction.
-
Some sections contain repetitive information. For example, in Sections 2.1 and 2.3, there is a repeated definition of public health. Similarly, Sections 2.5 and 2.6 include overlapping content that should be streamlined for clarity and conciseness.
-
The Synthesis of Key Trends section (4.2) needs to be expanded to provide a more in-depth analysis of the identified patterns and their implications.
-
The Conclusion section requires further development, particularly in elaborating on the study’s theoretical contributions and practical implications. Additionally, more focus should be placed on the existing literature gaps and concrete recommendations for future research. As it stands, this discussion needs to be substentially enalrged as it is now too brief, whereas it represents the main contribution of the study.
-
As I suggested in the previous review, the writing style would benefit from merging smaller paragraphs to create more cohesive and structured sections, particularly in the introduction. This would enhance readability and ensure a smoother flow of ideas.
-
Lastly, I had expected a more detailed response to my concerns from the last review round. Please provide a comprehensive report addressing all comments and revisions made in the next submission.
I wish the authors the best in revising their manuscript.
Author Response
- Theoretical framework: Explicitly state that these theories are the most cited in the analyzed literature and include a brief introduction in the introduction section.
Response:
We have revised the Introduction to incorporate a brief introduction to Legitimacy Theory and Stakeholder Theory as the primary theoretical perspectives framing our study. Additionally, we explicitly state that these theories are among the most cited in the analyzed literature, further justifying their inclusion.
Changes made:
- Added a paragraph in the Introduction summarizing both theories.
- Revised the Theoretical Framework section to clarify that these theories are highly cited in the field of CSR and community health.
- Some sections contain repetitive information (e.g., Sections 2.1 and 2.3, Sections 2.5 and 2.6).
Response:
We have streamlined these sections to remove redundancy. The definition of public health now appears only once in Section 2.1, eliminating repetition in Section 2.3. Similarly, overlapping content in Sections 2.5 and 2.6 has been consolidated to improve clarity and conciseness.
Changes made:
- Section 2.1: Maintains the definition of public health.
- Section 2.3: Removed repeated definition of public health.
- Sections 2.5 and 2.6: Merged overlapping content related to content analysis and bibliometric methods.
- Expand Section 4.2 (Synthesis of Key Trends) to provide a more in-depth analysis of the identified patterns and their implications.
Response:
We have expanded Section 4.2 to provide a deeper analysis of how the identified trends shape the research landscape. The revised section now explores:
- How CSR’s role in public health and sustainability has evolved over time.
- The increasing focus on industry-specific CSR efforts (e.g., food, alcohol, pharmaceuticals).
- Theoretical and practical implications of these shifts.
Changes made:
- Added further analysis on how these trends inform future CSR research.
- Discussed how CSR frameworks in public health have influenced corporate policies.
- The Conclusion section needs further development, particularly in elaborating on the study’s theoretical contributions and practical implications.
Response:
The Conclusion has been substantially expanded to:
- Provide a clearer discussion of theoretical contributions, including how our findings refine CSR theories in the context of public health.
- Offer concrete practical recommendations for policymakers, businesses, and researchers.
- Highlight existing literature gaps and suggest future research directions.
Changes made:
- Strengthened the discussion on CSR’s role in global health governance.
- Added practical applications of findings in corporate and policy contexts.
- Improve writing style by merging smaller paragraphs for better cohesion, particularly in the Introduction.
Response:
We have revised the Introduction by merging shorter paragraphs, ensuring a smoother flow of ideas. This improves readability and strengthens the logical progression of the arguments.
Changes made:
- Consolidated multiple paragraphs in the Introduction.
- Improved transitions between key sections.
- Provide a more detailed response to previous review comments.
Response:
We acknowledge that our previous response lacked depth. In this revision, we have provided comprehensive justifications for all modifications, ensuring transparency in our revision process. This response letter now details every change made in response to reviewer feedback.
Reviewer 4 Report
Comments and Suggestions for Authors
I believe the discussion needs an expansion, as you identify the trends, but what does this mean outside the bibliometric details and how can it impact the body of knowledge?
After the identified trends should be Theoretical Contributions, I would recommend looking at Homer, S. T., & Lim, W. M. (2024). Theory development in a globalized world: Bridging “doing as the romans do” with “understanding why the romans do it”. Global Business and Organizational Excellence, 43(3), 127-138. This will help you position your study and how you can develop the theory, there are many CSR theories, how does your study add or adapt them within the specific health community?
Following this, should be Practical Recommendations, here you should discuss what researchers can do with your results. Are there particular countries, contexts or themes that have a specific lack of focus and need more research doing? This may be more about looking for what is missing and why, rather than what has been researched extensively.
Author Response
- Expand the discussion section to go beyond bibliometric details and explain its broader impact on the body of knowledge.
Response:
We have significantly expanded the Discussion to:
- Explain how CSR’s impact on public health, policy-making, and corporate behavior extends beyond bibliometric trends.
- Connect bibliometric findings to real-world applications in industry and governance.
Changes made:
- Added a new subsection discussing how CSR strategies influence policy-making, consumer behavior, and regulatory frameworks.
- Include a Theoretical Contributions section and reference Homer & Lim (2024) to position the study within a broader CSR theoretical framework.
Response:
We have added a Theoretical Contributions subsection, incorporating insights from Homer & Lim (2024). This section discusses:
- How our study extends existing CSR theories by emphasizing public health as a central component.
- The evolution of CSR in response to global crises (e.g., COVID-19).
- Theoretical gaps in CSR models applied to community health.
Changes made:
- Added references to Homer & Lim (2024) to contextualize CSR theory development.
- Include a Practical Recommendations section highlighting areas needing further research and policy interventions.
Response:
We have added a Practical Recommendations subsection that:
- Identifies under-researched CSR topics in low-income countries and emerging markets.
- Suggests policy interventions to enhance CSR effectiveness in public health initiatives.
- Highlights the need for interdisciplinary approaches to CSR research.
Changes made:
- Added new recommendations tailored to researchers, policymakers, and corporations.